# Fire Size of Gasoline Pool Fires

**DOI:** 10.3390/ijerph17020411

**Published:** 2020-01-08

**Authors:** Iveta Marková, Jozef Lauko, Linda Makovická Osvaldová, Vladimír Mózer, Jozef Svetlík, Mikuláš Monoši, Michal Orinčák

**Affiliations:** 1Department of Fire Engineering, Faculty of Security Engineering, University of Žilina, Univerzitná 1, 010 26 Žilina, Slovakia; linda.makovicka@fbi.uniza.sk (L.M.O.); vladimir.mozer@fbi.uniza.sk (V.M.); jozef.svetlik@fbi.uniza.sk (J.S.); mikulas.monosi@fbi.uniza.sk (M.M.); michal.orincak@fbi.uniza.sk (M.O.); 2Slovnaft a.s., 824 12 Bratislava, Slovakia; jozef.lauko@slovnaft.sk

**Keywords:** gasoline, pool fires, mass burning rate, heat release rate

## Abstract

This article presents an experimental investigation of the flame characteristics of the gasoline pool fire. A series of experiments with different pool sizes and mixture contents were conducted to study the combustion behavior of pool fires in atmospheric conditions. The initial pool area of 0.25 m^2^, 0.66 m^2^, and 2.8 m^2^, the initial volume of fuel and time of burning process, and the initial gasoline thickness of 20 mm were determined in each experiment. The fire models are defined by the European standard EN 3 and were used to model fire of the class MB (model liquid fire for the fire area 0.25 m^2^), of the class 21B (model liquid fire for the fire area 0.66 m^2^), and 89B (model liquid fire for the fire area 2.8 m^2^). The fire models were used to class 21B and 89B for fuel by Standard EN 3. The flame geometrical characteristics were recorded by a CCD (charge-coupled device) digital camera. The results show turbulent flame with constant loss burning rate per area, different flame height, and different heat release rate. Regression rate increases linearly with increasing pans diameter. The results show a linear dependence of the HRR (heat release rate) depending on the fire area (average 2.6 times).

## 1. Introduction

Flammable liquids pose challenging problems for fire protection because they may ignite easily (depending on their flash/fire point), burn with high heat release rate (HRR), spread the flame rapidly, and are difficult to extinguish [1,2]. The fire spreads across the surface of the flammable liquid over its whole open area. Crude oil and its products have a high calorific value and flame temperature in which fire reaches up to temperature 1.400 °C [3,4]. The flammable liquids in the form of liquid fuels remain the dominant group of materials used in practice. The risk of spillage and subsequent ignition is always present.

Pool fires represent one of the most common accidental events which can occur within an industrial installation, with severe secondary effects such as re-ignition and spread to other flammable liquids nearby [5]. In the case of a liquid fuel fire, it is necessary to differentiate between fuel spill and pool. The pool fire is represented by a known area (for example an open container) with a limited volume of fuel. The fuel spill can be characterized as either continuously flowing or instantaneous [6].

The physical properties of flammable liquids support the burning process. Evaporation of liquids from their free surface (driven by the saturated vapor pressure above their free surface) produces vapors that mix with the air and form a flammable mixture (driven by the diffusion of vapor molecules with oxygen molecules). The process of burning liquids may be therefore characterized as the progression of the flame front through flammable vapors above the free surface of the liquid [3,7,8].

There are several types of scenarios for the investigation of liquid burning processes, and these are described by several authors [5,7,9,10,11]. Gasoline is one of the most frequently transported [12], stored, pumped, and used liquids in Slovakia and other European countries (Figure 1). The consequences of traffic accidents may involve discharge of gasoline onto the road and result in a pool fire or still fire. 

Pool fires are defined as flames established over horizontal fuel surfaces (as opposed to wall fires, which involve vertical fuel surfaces) [11]. Generally, pool fire’s surfaces have defined boundaries, and the depth of the liquid fuel pool is driven by the accumulation of fuel in the prescribed area. Gottuk and While [6] added one more minimum parameter—the depth of the liquid pool greater than 1 cm. 

The area (pool) of the initial body/volume of fuel will correlate to the size of the resulting fire [6,13]. Intensive research has been carried out over decades on this subject [7,13,14,15]. But, only a small proportion of the work has looked specifically at large-scale pool fires [15]. On the basis of heat release rate measurement and the observation of liquid fuel, Chen et al. [16] evaluated that the burning behavior of a pool fire can be divided into the following stages: initial growth, steady burning, boiling burning, and decay stages.

Characteristics of fire size for liquid pool fires are basic information to assess the behavior of spilled flammable liquids in case of their fire in the environment.

Numerous studies have described the different parameters controlling the behavior of pool fire (including fuels like gasoline, crude oil, kerosene, and ethanol). Steinhaus et al. [11] provided a detailed discussion of the defined “measurable quantities”, later “fire sizes” (first described by Drysdale [17]), of pool fires and different physical factors affecting the behavior of large pool fires. The main “measurable quantities” associated to a pool fire, by Steinhaus et al. [11] and Drysdale [17], are “Burning rate” or “Mass loss rate” in (kg s^−1^), Heat release rate (*HRR*) in (kW) or (MW), Flame height (m), Flame temperature (°C), Smoke production rate in (m^3^ s^−1^) or (kg s^−1^), and Radiation. Radiation is described either as the emissive power at a given point in space (kW m^−2^) or as the sum of all heat lost by radiation (kW), the latter often expressed as a percentage of the *HRR* [11,18]. 

The physical characteristics of the pool fire can be summarized following Steinhaus et al. [11]: Pool geometry (diameter, depth, substrate), Fuel composition, Ventilation conditions (wind, forced, or restricted ventilation, etc.), Surrounding geometry (open air, compartment height, proximity to walls, thickness of layer, etc.), and Nature of the bounding materials, that is, those used to construct the walls of liquid pool fire vessels. Zhao et al. [19] investigated the influence of the thin-layer burning behavior of gasoline, including the heat flux feedback to the burning surface, the penetrating thermal radiation, the temperature profile of liquid layer, and the burning rate, which were studied in experiments with thin-layer pool fires in square, fireproof glass trays.

The fire size is primarily characterized by the heat release rate (HRR) and flame height [6]. The burning rate has been expressed in terms of a “regression rate”, given in mm·min^−1^. This parameter presents the liquid surface descent by a number of mm per minute as the fuel is consumed in the fire. The burning rate per unit area (and hence most other characteristics) of a pool fire increases with pan diameter up to about 2–3 m [11]).

The theory developed by Babrauskas [20] is accepted [5,6,9,10,11,14,21,22] as a basis for assessing the heat release of pool fires. Babrauskas [20] laid the foundations for the calculation of the “fire size” of flammable liquids, if the combustion mode is specified (Table 1).

The *HRR* of the fire is usually determined by tests at various scales (mid- to large-scale tests). The obtained experimental data are the basis for calculation of *HRR* for the fire, given by the following equation [23]: (1)$$Q=m^{\prime \prime }\;\Delta H_{c,eff}\left(1-e^{-k\beta D}\right)$$
where *m*″ is the burning rate or mass loss rate per unit area per unit time (kg m^−2^ s^−1^), ∆*H_c,eff_* is the effective heat of combustion (kJ·kg^−1^), *kβ* is the empirical constant (m^−1^), and *D* is the diameter of burning area (m). Babrauskas [20] also indicates of values *m*″ and *kβ* for most common liquid fuels. For noncircular pools, the effective diameter is defined as the diameter of a circular pool with an area equal to the actual pool area, given by the following equation:(2)$$D=\surd \frac{4A}{\pi }$$

The flame height is a dynamic parameter, and the flame tip is often taken to be the point of 50% intermittency [24], because the visual effect of the flame is not entirely representative of its height. The best known and most widely adopted correlation for calculating the ratio between the flame height and the diameter of a circular pool is described by Bubbico, Dusserre, and Mazzarotta [5], Chen and Wei [10], and Thomas [25]: (3)$$\frac{H}{D}=42\;{\left(\frac{m^{\prime \prime }}{\rho_{a}\surd gD}\right)}^{0.61}$$
where *ρ_a_* is the air density (kg m^−3^), and *g* the gravitational acceleration (m s^−2^).

Heskestad [26] has analyzed experiment results from a wide variety of sources including pool fires and derived the follow relation: (4)$$\frac{H}{D}=0.235\;\frac{Q^{\frac{2}{5}}}{D}-1.02$$
where *H* is Flame height (m), *D* is diameter (m), and *Q* is the heat release rate HRR (kW).

Zhang et al. [27] represents, by the B_r_L_tz_ correlation, an even simpler approach, since the height of the flame is dependent on the diameter of the pool only: (5)$$\frac{H}{D}=1.73+0.33\;D^{-1.43}$$

Lam and Weckman [28] and Gialdi Salvagni, Centenoand Sperb Indrusiak [29] offered an overview of calculation flame height formulas.

The aim of the article is to specify the behavior of the oil product Petrol Super 95 (BS95) (nonpolar liquid) when ignited. The burning process was investigated through a pool fire scenario defined as fire model Class B by European standard EN 3-7 [30]. This study aims to get a better understanding of the (heat) requirements for the pool fire of the actual fossil fuels in terms of their thermal properties. The role of research is to measure the dependence of the development of pool fires and fire sizes on the average pan and burning time.

## 2. Materials and Methods 

### 2.1. Fire Model Class B

Gasoline as fuel is usually stored in large capacity tanks, transported in tankers or smaller containers, and pumped into vehicles. Pool fire is, therefore, a representative model of a real gasoline fire resulting from a spill or discharge. Gasoline “Petrol Super 95 (BS95)” was obtained from Slvonaft. At the time of the experiment, gasoline contained 10% ETBE (ethoxy-2-methylpropane content). All fuel details are given in the Material Safety Data Sheet of gasoline [31].

The experiments were done in tested pans (Table 2 and Figure 2). The basis for the preparation of the test pans was chosen from relevant European standards and modified according to the conditions of EN 3-7 [30]. The models have been used for multiple purposes.

Class B fire models have been prepared in three variations, as the Fire Model Class MB (represents a standard 200 L metallic barrel for the transportation of liquid products), Fire Model Class 21B, and Fire Model Class 89B. The choice of pan diameter *D* was associated with various burning modes and regimes (see Table 1).

The impact of fuel layer depth on heat release rate with a pool fire in the convective, turbulent burning mode (i.e., 0.05 m ≥ *D* ≥ 0.2 m) is detail described by Babrauskas [20]. During the experiments, only the turbulent flame was observed.

### 2.2. Test Conditions and Behaviour of Experiments

The tests were made under atmospheric conditions; the ambient temperature ranged from 0 °C to 20 °C, the wind speed up to 3 m s^−1^. The air flow rate in any given experiment did not exceed 3 m s^−1^ (a prerequisite for the implementation of Class A and Class B fire models according to EN 3-7 + A1 [30]).

Each test pan was filled with the initial fuel mixture to achieve a pool depth of 20 mm. The amount of burned fuel and burning time was determined by initial indicative tests. The bottom of the tested pan, after the experiment, had been covered with a continuous layer of flammable liquid, for abide would need keep the fire on the whole pan. 

The experiment was done by firefighters of Rescue Corp of Slovnaft a.s. in Bratislava, Slovakia, where whole security conditions and occupational health and safety are observed. The whole process was recorded on digital video combined with photo documentation using two different indicators. The flame geometrical characteristics were recorded by a CCD (charge-coupled device) digital camera.

Each of the experiments was repeated three times. As expected, the experiments were accompanied by the production of large quantities of dense black smoke, the properties of which were not monitored.

The experiments were performed for the burning duration of 1, 3, and 5 min. The fire was extinguished by covering it with a metal hatch after the desired burning time. 

To obtain the same starting conditions for all experiments, the pans were left to cool down to the ambient temperature following each fire test.

Finally, wet fabrics were used to isolate liquid from the air. At the same time, the walls of the tank and the hatch were cooled by a stream of water to prevent deformation of pan and damage to the cover fabric. The schematic diagram of the experimental setup is shown in Figure 2.

The Fire Model Class 89B fire model with a surface area of 2.8 m^2^ was additionally modified. The realization of this large-scale experiment resulted in a continuous burn exceeding the defined period. This was due to ferocious burning which could not be fully interrupted. This experiment was terminated by fuel burnout. The burning time was watched. After orientation experiments, the research time was set to 140 s.

Based on the set burning time on the given fire area, fire sizes such as heat release rate (HRR), burning regression rate, mass burning rate per unit, and flame height were determined and compared. The obtained results were evaluated by descriptive statistics and presented average value and standard deviation.

### 2.3. Experimental Methods for Monitoring of Fuel Quality before and after a Model Fire

Gasoline test samples were analyzed by fuel quality assurance tests. The gas chromatography is used in the application of simulated distillation techniques [32,33,34,35,36]. This fact is used to monitor the quality of petroleum products.

The quality of petroleum products was evaluated by the density parameters (done at 15 °C, according to [37]), determination of atmospheric pressure distillation characteristics, and determination of hydrocarbon types and oxygenates in gasoline by multidimensional gas chromatography method according to [38].

Experiments were done to compare the quality of the gasoline before and after the fire (sampling the rest of the burnt fuel).

## 3. Results and Discussion 

### 3.1. Results for Fire Size

Fossil fuels, like organic substances, burn with a flame, which depends on different conditions. Muños at al. [39] dealt in detail with the shape and character of the fossil fuel flame. Gasoline is a complex chemical mixture [40] of hydrocarbons consisting primarily of paraffins, cycloparaffins, and aromatic and olefins hydrocarbons having carbon numbers predominantly higher than C3 (Table 3) [31].

The fuel consumption results over time are generally known (Table 2). But, by increasing the burning time, the regression rate increases linearly (Figure 3), however, the mass burning rate per unit area for each burning time was confirmed [20]. At the same time, the linear dependence of the mass burning rate on the area of fire is confirmed (Table 2).

As can be seen with the increase in the area of fire, the firing of the fire model class 21B is more intense in the area of 0.66 m^2^, at the same time as in the case of the fire model MB class fire on 0.25 m^2^. The results obtained do not offer the possibility of extrapolating the increase or decrease in the linear dependence of the amount of fuel burned away from the fire area.

Mass burning rates of organic liquids were investigated by Gottuk a White [6]. Value of *m*″ between 0.045 and 0.64 kg m^−2^ s^−1^ for pool fire of gasoline spill fire was 0.01 l with diameters greater than 1.5 m. Mass burning rate of gasoline pool fires decreased with *D* = 1.8 m. This results corresponds with Bubbico, Dusserre, and Mazzarotta [5] results for *D* = 2 m. 

Burning rates of organic liquids were investigated by Gottuk and White [6]. The value *m*″ is in the range of 0.045–0.64 kg m^−2^ s^−1^ for pool fire, and *m*″ for the gasoline spill fire was 0.011 kg m^−2^ s^−1^ for diameters greater than 1.5 m. The mass burning rate of gasoline pool fires decreased with *D* = 1.8 m. The results correspond with Bubbico, Dusserre, and Mazzarotta [5] results for *D* = 2 m.

Niass et al. [41] conducted a series of experiments where he tested flammable fuel mixtures enriched with oxygen due to blends formed with alcohols and ETBE (ethoxy-2-methylpropane content). The result of the research confirmed the suitability of using ETBE as a MON (mark for “motorová nafta” = diesel fuel) petrol spraying agent.

The dependence of *HRR* on the area of fire *A* is linear (Figure 4). Perfection is in the same amount of heat released in different time intervals at the same pan’s diameter. The *HRR* dependence on the diameter is linear with the statistical evaluation R^2^ = 0.9735 for the burning time of 60, 180, and 300 s. The HRR does not depend on burning time, but depends on the fire area (Figure 4).

Xiao et al. [42] carried out experiments with pool fires of methanol, gasoline, and diesel to evaluate different types of fuel. The diameter and height of the fuel pan are 0.42 m and 0.03 m, respectively, for demonstration of small and large expanded fire size. The purpose of these experiments was to measure the radiation heat flux as a parameter of HRR. And the free-burning period was followed by fire extinction by nitrogen, which was the main focus of the study.

As is shown in Figure 5a, the first five pictures indicate fire expansion. These pictures present the typical characteristics of flame height fluctuations (Figure 5). Lin et al. [43] and Zhang et al. [44] investigated the influence of horizontal cross flows on the flame geometry in pool fires. Gottuk and White [6] presented graphic relation between regression rate and pan diameter, where laminar flow regime is observed only up to 1 m. A limit of 1 to 2 m is presented as transition flow regime. Our observations were only of turbulent flow regime. 

The fire size depends on the fuel type, liquid pool size, fuel thickness, and other factors [5,23,45,46,47,48]. In this paper, the flame height for various pool diameters is calculated by formulae in Equations (3)–(5). The results are shown in Figure 6. The Organization of United States Regulatory Commission prepared software [49] on calculation of fire size, and we can compare our experimental results and calculated results. One entered parameter was different—density. Because, our experimental samples of gasoline had measurement value 780 kg m^−3^, and Software formula offers only 740 kg m^−3^ value. The different results are presented on Figure 6.

### 3.2. Results of Monitoring the Change in Gasoline Quality after Pool Fire

Preliminary experiments—indicative—initial measurements with randomly selected samples with the objective of identifying changes in the quality of test samples.

A gasoline sample, labelled B0, was collected prior to the experiment, and a BP gas sample was obtained after the pool fire, which was the burning residue (Table 3).

Gasoline density increases after pool fire (Table 3). This fact is supported by the presence of 150 ppm of water in the original B0 sample, and in the BP sample, the water could not be determined.

Determination of distillation characteristics at atmospheric pressure of samples B0 and BP shows a change in gasoline quality.

The quality measurements of the analyzed samples were repeated with the exact sample specification. The first measurements were made on sample B, where we took out the test amount of fuel before the experiment (labelled as B) and after each experiment observing “fire size”, so after 60, 140, 180 and 300 s (Figure 7). The resulting samples, designated B1 (after 60 s pool fire), B2 (after 140 s fire), B3 (after 180 s fire), and B4 (after 300 s pool fire) were analyzed as B0 and BP samples. In the BP sample, the two layers identified were immiscible with each other. It is difficult to assume which changes are involved, as the fractions in the BP sample are at the beginning higher and in the last phase of distillation are lower. However, due to the formation of immiscible fractions, the absence of (loss of) or release/burning of the fractions of the chemical providing the miscibility of the polar and nonpolar fractions in the gasoline (i.e., the proportion of pure gasoline and bio-component) has arisen. Multidimensional gas chromatography (Table 3, Figure 8) also confirms the change in original sample quality after burning on a sample of BP.

Additives fractions are increasing, which means that they were not subject to evaporation and combustion as much as other fractions, which decreased in comparison to the unburned sample.

The Mealy et al.’s [48] distillation curves for the gasoline, diesel, and kerosene measured in accordance with ASTM D287 [38] had the same shape as in our experiences. The changes occurred in the temperatures values. Distillation points 0.8, 0.9, and 1 were, in our observations, higher than in the Mealy et al.’s [48] distillation curves. The differences were likely the consequence of different pool fire experiments. By continuing to search for the reasons for the changes, the graphical dependence of the variation of the distillation ratios in the samples examined was shown (Figure 7). 

As can be seen, with the increasing fire time, distillation fractions increase. This claim is consistent with the system of composition of gasoline and the combustion of individual gasoline fractions according to their physical parameters.

Determination of hydrocarbon types and oxygenates in gasoline by multidimensional gas chromatography method of samples B, B1, B2, B3, and B4 to the first and second distillation stages indicated a gradual burn-out of these components, which was reflected in a decreasing shape of acquired dependence (Figure 8).

## 4. Conclusions

The presented research is a contribution to the growing body of knowledge regarding the burning behavior of gasoline—a highly flammable liquid present in almost all fields of industry and daily life. The focus of this work was two-fold:Investigate the burning behavior of gasoline pool fires primarily through visual monitoring of flame parameters and calculation of other parameters by employing available correlations.Investigate the impact of combustion in the pool fire configuration on the chemical composition of the fuel samples.

The following conclusions may be derived from the experiments carried out:For the investigated pan diameters (0.56–1.89 m), immediate flame spread of flame was observed. Hence, sufficiently rich fuel-vapor/air mixture layer above the liquid surface was formed, which may be attributed to the saturated vapor gas pressure and low flash/fire points of the investigated fuel.Dependence of *HRR* is directly proportional to the area of fire in the investigated range. The *HRR* dependence of the area of fire is linear with the statistical evaluation *R**^2^*** = 0.9735 for the burning time of 60, 180, and 300 s. The HRR does not depend on burning time.Significant inconsistencies were found in the observed and calculated flame height. The flames expressed highly turbulent behavior during the experiment, which was strengthened by wind. Measured flame heights were in the interval 2–4.63 m, depending on the pan diameter. The Thomson, Chen and Wei, and Zheng flame height calculation results were different, occupying data within ±2.5–5 m.The combustion process affects the composition of the fuel by two mechanisms. Elevated temperatures cause the lighter fractions to evaporate and contribute to combustion in the earlier phases of fire. Secondly, the combustion process as a chemical reaction is not ideal, hence the products of incomplete combustion mix with the original fuel fractions, leading to the changes indicated in the results of the multidimensional gas chromatography.

## Figures and Tables

**Figure 1 ijerph-17-00411-f001:**
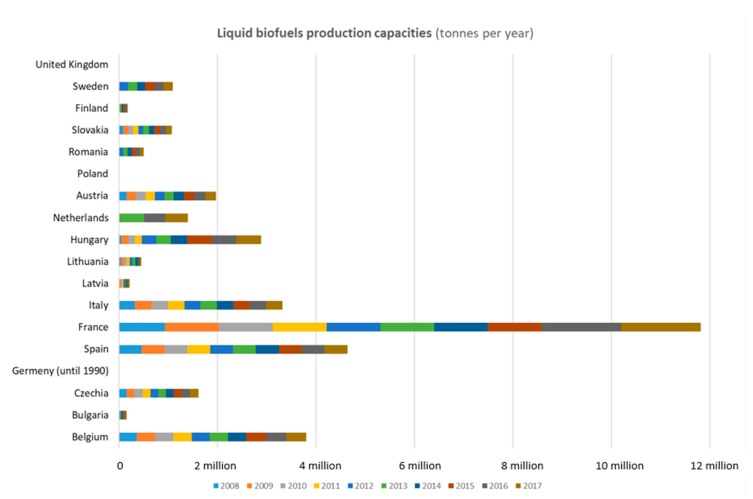
Liquid biofuels production capacities (Source: Eurostat. Last update: 25 February 2019).

**Figure 2 ijerph-17-00411-f002:**
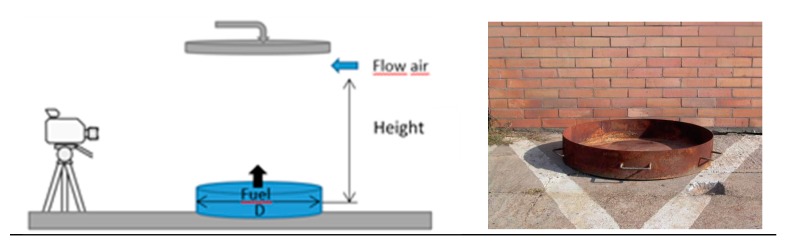
The experimental setup of pool fire and real picture.

**Figure 3 ijerph-17-00411-f003:**
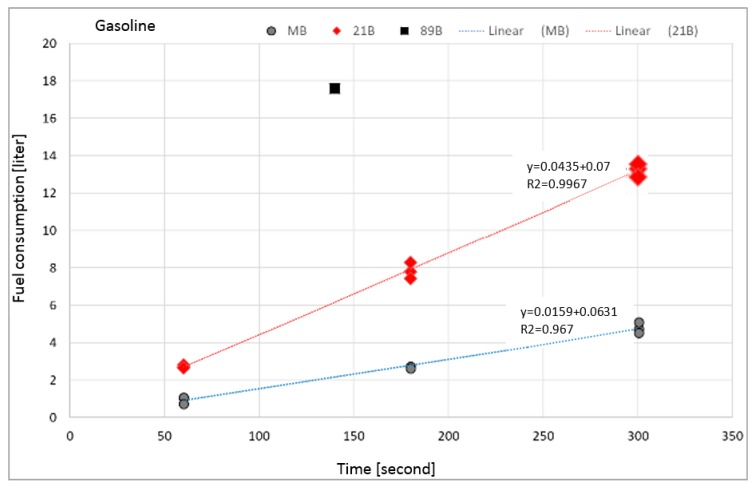
Variation of fuel consumption as a function of time.

**Figure 4 ijerph-17-00411-f004:**
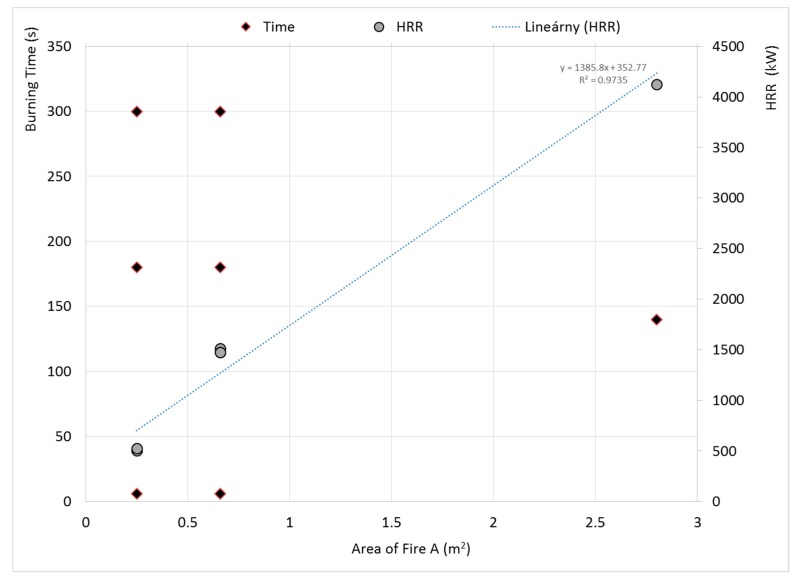
Relation between HRR from Area of Fire.

**Figure 5 ijerph-17-00411-f005:**
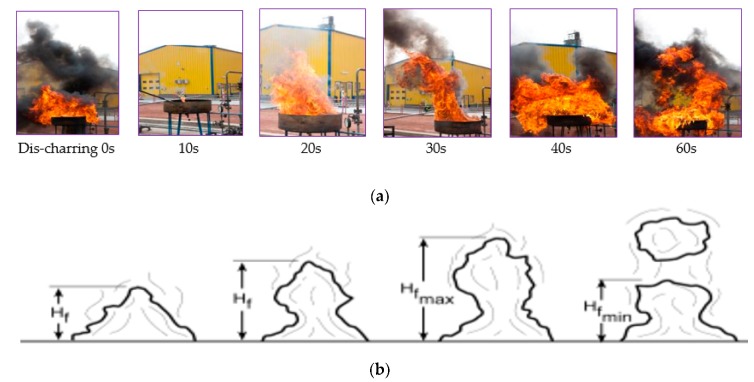
Flames of gasoline pool fire. (**a**) Typical pictures of fire expansion phenomenon of gasoline pool fire. (**b**) Characteristics of Flame Height Fluctuations [23].

**Figure 6 ijerph-17-00411-f006:**
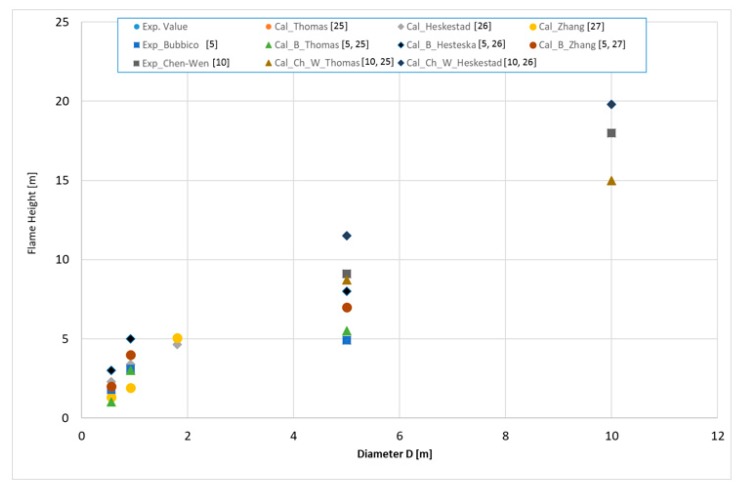
Flame height for gasoline. Experimental data and calculated predictions. Legends: Exp.—presents experimental values, Cal presents calculated values by actual author.

**Figure 7 ijerph-17-00411-f007:**
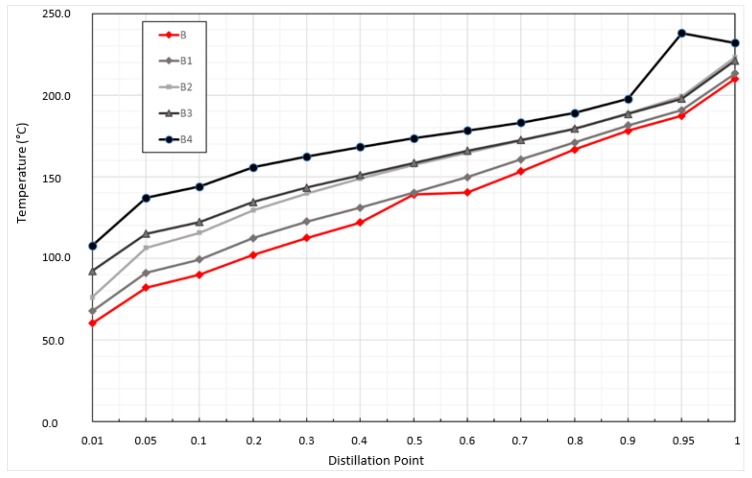
Distillation curves for the gasoline samples, measured in accordance with Methodology ASTM D287/ISO 3405: 2011 [38].

**Figure 8 ijerph-17-00411-f008:**
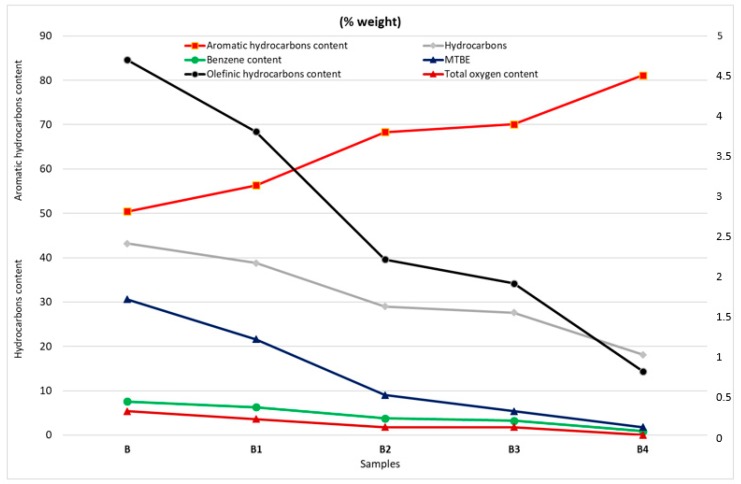
Multidimensional gas chromatography of samples B, B1, B2, B3, and B4.

**Table 1 ijerph-17-00411-t001:** Determination of small or large pool fire on basic *HRR* by selected authors.

Diameter D (m)	Babrauskas [20]	Steinhaus et al. [11]	Test Ref.
	Burning Mode	Burning Regime	
<0.05	Convective, laminar	Small pool fire: convectively dominated burning for pools	-
0.05 to 0.2	Convective, turbulent		
0.2 to 1.0	Radiative, optically thin	Large pool Fire: radiatively dominated burning for pools	MB, 21B
>1.0	Radiative, optically thick		89B

Legend: MB, 21B, and 89B are tested variant fires in this article.

**Table 2 ijerph-17-00411-t002:** Class B fire models, fuel, gasoline Super 95.

Conditions of Pool Fire	MB			21B			89B
	EN 1568-4			EN 3-7 + A1			
A (m^2^)	0.25			0.66			2.8
D (m)	0.56			0.92			1.89
Air Temperature (°C)	10.8			11.5			11.8
Wind (m·s^−1^)	1.55			2.2			2.2
Burning time (s)	6	180	300	6	180	300	140
Volume of fuel (l): 1.5 L H_2_O + BS 95	3	4.5	6	6	12	15	17.6
Fuel consumption (L s^−1^)	0.0156 ± 0.003	0.0149 ± 0.003	0.0157 ± 0.001	0.0452 ± 0.001	0.0441 ± 0.002	0.0442 ± 0.001	0.123 ± 0.002
H experimental (m)	2	2.2	2.2	3.41	3.41	3.41	4.63
Fire Size							
*m’* (kg s^−1^)	0.0122 ± 0.002	0.0116 ± 0.002	0.0122 ± 0.001	0.0349 ± 0.002	0.0349 ± 0.002	0.0333 ± 0.001	0.0877 ± 0.006
*m”* (kg m^−2^ s^−1^)	0.0490 ± 0.008	0.0463 ± 0.001	0.0484 ± 0.003	0.0524 ± 0.002	0.0511 ± 0.003	0.0512 ± 0.001	0.0341 ± 0.001
*HRR* (kW)	510	506	528	1511	1474	1477	4127

**Table 3 ijerph-17-00411-t003:** Physical and chemical properties of investigated Gasoline (MSDS of gasoline [31]) and comparison of selected shares in B0 and BP and in gasoline samples.

Physical and Chemical Properties	MSDS for Gasoline [31]	Monitored Shares
B0	BP
Density at 15 °C (kg m^−3^)	780	789.4	812.9
∆H_c,eff_ (kJ kg^−1^) by Babrauskas [20]	43.700		
Distillation range (°C)	30–260	50–250	70–210
Aromatic hydrocarbons content (% weight)	35.04	47.9	58.7
Paraffins content (% weight)	31.99	44.9	36.6
Benzene content (% weight)	1.04	0.42	0.32
Methyl tert-butyl ether MTBE (% weight)		1.9	1.0
Total oxygen content (% weight)		0.3	0.2
Olefinic hydrocarbons content (% weight)	15.5	5.3	3.8
Toluene content (% weight)	7.48		
Naphthenic hydrocarbons content (% weight)	7.46		
n-Hexane content (% weight)	1.49		
2-ethoxy-2-methylpropane content (% weight)	≤15		
Ethanol content (% weight)	≤5		
Diisobutylene content (% weight)	app 0.09

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
