# Peer review of "Fire Size of Gasoline Pool Fires"

_ijerph, 2020, doi:10.3390/ijerph17020411_

Round 1

Reviewer 1 Report

Specific comments:

-- Equation (3) is not correct (an exponent is lacking)

-- Fig. 2: “Weight” should rather be “Height”.

-- Section 3.1, first line: The authors say “Fossil fuels… burn with a luminous flame (Figure 6).”

However, liquid fossil fuels are characterized by a rather bad combustion, and the flame has usually part of its surface covered by black smoke. This can be seen in Fig 5 (not 6), where very luminous zones exist together les luminous ones.

It should be taken into account that the paper deals with very small pool fires; with larger pool fires (for ex., D = 5 m) this phenomenon is much more important.

-- Figure 3 footnote: “Relation between view factor fuel consumption from time”.

“View factor2? View factos has nothing to do with this figure!

-- Figure 4. This trend seems not logical at all. HRR is proportional to the burning velocity in the pool fire; and burning velocity, at steady state, is proportional to the pool surface, i. e. to D2, not to D.

-- Figure 5: in some cases, the shape, height and length of the flames is affected by the wind. How have the authors managed this?

-- Fig. 6: Why the authors compare the different published equations at diameters up to 10m, if their experimental results are limited to D = 1.89 m ?

Author Response

We would like to thank the opponent for the valuable comments. We have tried to carefully and consistently modify the article to improve its quality and offer our experimental results. We have the following comments for each request:

-- Equation (3) is not correct (an exponent is lacking). 

Thank you for the notification. It is corrected.

-- Fig. 2: “Weight” should rather be “Height

Thank you for the notification. It is corrected.

-- Section 3.1, first line: The authors say “Fossil fuels… burn with a luminous flame (Figure 6).”

However, liquid fossil fuels are characterized by a rather bad combustion, and the flame has usually part of its surface covered by black smoke. This can be seen in Fig 5 (not 6), where very luminous zones exist together les luminous ones.

It should be taken into account that the paper deals with very small pool fires; with larger pool fires (for ex., D = 5 m) this phenomenon is much more important.

-- Figure 3 footnote: “Relation between view factor fuel consumption from time”.

“View factor2? View factos has nothing to do with this figure! 

Apologies, it was incorrect translation.

-- Figure 4. This trend seems not logical at all. HRR is proportional to the burning velocity in the pool fire; and burning velocity, at steady state, is proportional to the pool surface, i. e. to D2, not to D.

The statement is correct. The selected parameters in FIG. 4 are easy to read. With the change of the "x" axis from D (m) to A (m2) the results were confirmed  We just added to our statement: Lines 210-211 The HRR does not depend on burning time, but depends on the fire area (indicated during parameter D (m) in Figure 4).

-- Figure 5: in some cases, the shape, height and length of the flames is affected by the wind. How have the authors managed this.

Table 2 shows the experimental conditions with wind velocities values from 1.55 m.s-1 for the fire area MB and 2.2 m.s-1 for 21B and 89B. The experiments were carried out deliberately at the wall (leeward) as seen in FIG. 2 and the microclimatic conditions were monitored to carry out the experiment under the same conditions.

-- Fig. 6: Why the authors compare the different published equations at diameters up to 10m, if their experimental results are limited to D = 1.89 m ? 

We used available literature. Sorry, we were unable to find results comparable to our experimental conditions.

Reviewer 2 Report

The manuscript addresses an experimental investigation related to pool fire. The main problem is that it brings no novelty whatsoever. The writing is poor and the objective and challenge of the research is not stated. The reader really has a hard time as it seems as if the author just dump a bunch of results which do not bring any new information or knowledge for the scientific community.

In my opinion this material looks more like a undergraduate course work. Therefore as it is, the manuscript is well below the level of any descent journal.

Author Response

Based on the comment received, we are unable to provide any reply. The comment doesn’t bring up any specific question neither any relevant argument to react to. Despite everybody has a right to their opinion, we do not understand why there is a need to dishonest work of fellow researcher.

Author Response

We would like to thank the opponent for the valuable comments. We have tried to carefully and consistently modify the article to improve its quality and offer our experimental results. We have the following comments for each request:

We have adjusted the abstract The main objectives are presented at the end of the introduction. We merged the introduction into one whole and tried to complete it with logical continuity. Thank you for your feedback. Fuel information was added. Thank you for your comment. Thank you for your comment. Thank you for your comment. Fig. 4 has been adjusted according to the requirements of another reviewer. Thank you for your comment. It was reviewed. Thank you for your comment. It was reviewed. Thank you for your comment. It was reviewed.

10., 11. a 12. Thank you for your comment. It was reviewed.

Round 2

Reviewer 1 Report

As a general comment, I would say that the paper should be improved (I have the feeling that the authors should get more information on what has been done/published in this field).

As for the conclusions, the first three paragraphs should be deleted (they are not conclusions) and they should focuss on the interpretation of the results; a mention of wind effect (almost always present in real cases, as well as in the  two of the pictures of Fig. 5) should be introduced .

Specific comments:

-- Section 3.1, first line: "Fossil fuel, like organic substances, burn with a luminous flame" No! This is not true. The fires have luminous zones but also zones where the fire is covered by black smoke; this effect -what can be partly seen in Fig. 5) is more and more important in large fires (which are the ones usually found in real accidents).

-- Fig. 3 footnote should be "Variation of fuel consumption as a function of time".

-- Paragraph before Fig. 4: "The dependence of HRR on the pan diameter is linear (Fig. 4), with statistical evaluation 10.9735"  No! This is not true! What is true is that HRR varies linearly with the pan area. So: HRR = kD is not true, HRR = kA is true.

-- Fig. 6: more data should be analyzed (see, for ex., Muñoz et al., Combustion and Flame  139 (2004) 263-277).

-- Conclusion 2. Again the authors say "The HRR dependence on the diameter is linear with the statistical evaluation R2 = 0.9735". This is not true.

Author Response

We would like to thank the opponent for the valuable comments. We have tried to carefully and consistently modify the article to improve its quality and offer our experimental results and we and we received  the reviewer's comments.

Reviewer 2 Report

As previously pointed out the work brings no novelty that justify the publication. 

Author Response

Based on the comment received, we are unable to provide any reply. The comment doesn’t bring up any specific question neither any relevant argument to react to. Despite everybody has a right to their opinion, we do not understand why there is a need to dishonest work of fellow researcher.

We bring an experimental methodology and results, that can become an inspiration for further research.